# Increased public health threat of avian-origin H3N2 influenza virus caused by its evolution in dogs

**Mingyue Chen[1†], Yanli Lyu[1,2†], Fan Wu[1,2], Ying Zhang[3], Hongkui Li[4], Rui Wang[1], Yang Liu[2], Xinyu Yang[1], Liwei Zhou[1,2], Ming Zhang[5], Qi Tong[1], Honglei Sun[1], Juan Pu[1], Jinhua Liu[1], Yipeng Sun[1]\***

[1]National Key Laboratory of Veterinary Public Health Security, Key Laboratory for Prevention and Control of Avian Influenza and Other Major Poultry Diseases and Key Laboratory of Animal Epidemiology of Ministry of Agriculture and Rural Affairs, College of Veterinary Medicine, China Agricultural University, Beijing, China; [2]Veterinary Teaching Hospital, China Agricultural University, Beijing, China; [3]Department of Laboratory Medicine, the First Medical Centre, Chinese People's Liberation Army (PLA) General Hospital, Beijing, China; [4]Liaoning Agricultural Development Service Center, Shenyang, China; [5]Department of Epidemiology and Biostatistics, University of Georgia, Athens, United States

**\*For correspondence:**
sypcau@163.com

[†]These authors contributed equally to this work

**Competing interest:** The authors declare that no competing interests exist.

**Abstract** Influenza A viruses in animal reservoirs repeatedly cross species barriers to infect humans. Dogs are the closest companion animals to humans, but the role of dogs in the ecology of influenza viruses is unclear. H3N2 avian influenza viruses were transmitted to dogs around 2006 and have formed stable lineages. The long-term epidemic of avian-origin H3N2 virus in canines offers the best models to investigate the effect of dogs on the evolution of influenza viruses. Here, we carried out a systematic and comparative identification of the biological characteristics of H3N2 canine influenza viruses (CIVs) isolated worldwide over 10 years. We found that, during adaptation in dogs, H3N2 CIVs became able to recognize the human-like SAα2,6-Gal receptor, showed gradually increased hemagglutination (HA) acid stability and replication ability in human airway epithelial cells, and acquired a 100% transmission rate via respiratory droplets in a ferret model. We also found that human populations lack immunity to H3N2 CIVs, and even preexisting immunity derived from the present human seasonal influenza viruses cannot provide protection against H3N2 CIVs. Our results showed that canines may serve as intermediates for the adaptation of avian influenza viruses to humans. Continuous surveillance coordinated with risk assessment for CIVs is necessary.

## Editor's evaluation

This paper focuses on the avian H3N2 influenza virus that has recently started infecting and spreading between dogs. Using exhaustive and impressive experimental approaches, the authors demonstrate how this virus is adapting to dogs over time, gaining more and more properties consistent with robust infection of mammals. This paper is destined to become part of the canon on emerging viruses.

## Introduction

In the 21st century, newly emerging viruses, such as influenza A H7N9, Ebola virus, Zika virus, and the SARS-CoV-2 virus, are posing serious challenges to healthcare systems (*Jacob et al., 2020*; *Imai et al.,*

*2017*; *Yakob and Walker, 2016*; *Thakur and Ratho, 2022*). These challenges are constant reminders to the scientific community to pay attention to emerging animal-borne zoonotic diseases. Influenza A viruses have a relatively broad host range (*Long et al., 2019*). When animal-borne viruses with different antigenicities acquire human-human aerosol transmission abilities, they become epidemic in the population. The four human pandemic viruses in recorded history underwent avian or swine influenza virus gene reassortment with human influenza virus or acquired human adaptive mutations (*Vijaykrishna et al., 2010*). Animal-borne virus adaption to a mammalian intermediate host is an important way that they are able to establish infections in humans (*Parrish et al., 2015*). Swine are considered a typical intermediate host; for example, the Eurasian avian-origin lineage of H1 subtype swine influenza viruses that originated in European swine in the 1970s gradually accumulated amino-acid mutations related to human adaptation and gained increased infectivity in humans (*Mena et al., 2016*; *Brown, 2013*). However, the role of other mammals in viral ecology is still unclear.

Similar to that of pigs, the canine respiratory tract contains both types of sialic acid receptors used by influenza viruses (α2,3- and α2,6-linked) (*Wasik et al., 2017*; *Ning et al., 2012*). Dogs are susceptible to natural influenza virus infections caused by transmission from avian (H3N2 and H5N1), equine (H3N8), or human (pdmH1N1 and H3N2) virus reservoirs (*Lin et al., 2012a*; *Crawford et al., 2005*; *Song et al., 2008*). Several reassortment events in influenza viruses have occurred from different host sources in dogs (*Lee et al., 2016b*; *Voorhees et al., 2017*), such as reassortment between canine H3N2 and human H1N1 viruses (*Song et al., 2012*; *Moon et al., 2015*) and canine H3N2 and swine influenza viruses (*Eichorst et al., 2018*). Dogs are important companion animals, and once a new zoonotic disease appears in dogs, there is a high chance to infect humans. However, whether dogs can act as intermediate hosts to produce zoonotic influenza viruses is not established.

Although dogs have been found infected with multiple influenza viruses, only equine-origin H3N8 and avian-origin H3N2 viruses have established lineages in dogs (*Hayward et al., 2010*; *Anderson et al., 2012*; *Zhu et al., 2015*; *Lyu et al., 2019*). Compared with H3N8 CIV, H3N2 CIV has a broader host range, infecting multiple mammalian animals, including ferrets, guinea pigs, mice, and cats (*Lee et al., 2013*, *Lyoo et al., 2015*; *Jeoung et al., 2013*; *Song et al., 2011*). H3N2 CIV was first isolated in 2006 from Guangdong Province in China, and was found to be genetically most closely related to the H3N2 avian influenza viruses prevalent in aquatic birds in South Korea for all eight gene segments (*Li et al., 2010*; *Su et al., 2012*). Since then, H3N2 CIV has been prevalent in China (*Sun et al., 2013*; *Yang et al., 2014*; *Lin et al., 2012b*; *Wu et al., 2021*) and South Korea (*Lee et al., 2016a*), and has circulated in the United States since 2015 (*Voorhees et al., 2017*; *Voorhees et al., 2018*; *Martinez-Sobrido et al., 2020*; *Dalziel et al., 2014*). The long-term epidemic of avian-origin H3N2 virus in canines offers the best opportunities to investigate the potential role of dogs in the ecology of influenza A viruses.

In the present study, we thus systematically investigated the evolution of genetic and biological properties of this avian-origin virus during its circulation in dogs. We found that during the adaptation of H3N2 CIVs to dogs, H3N2 CIVs became able to recognize the human-like SAα2, 6Gal receptor, showed gradually increased HA acid stability and replication ability in human airway epithelial cells, and had a 100% transmission rate via respiratory droplets in a ferret mode. Our results revealed that dogs might serve as potential intermediate hosts for animal influenza viruses' adaption to humans.

## Results

### Continued genetic evolution of avian-origin H3N2 CIVs in dogs

From 2012 to 2019, we collected tracheal swab samples from 4174 dogs with signs of respiratory disease from animal hospitals and kennels in nine provinces or municipalities of China (*Figure 1—figure supplement 1*). A total of 235 samples (5.63%) were positive for H3N2 infection. The mean positive rates for each year increased from 1.98% in 2012 to 10.85% in 2019 (*Figure 1—figure supplement 2*), with a sharp increase after 2016. According to isolation time and location, 117 representative viruses were selected for full genome sequencing, including 51 strains isolated from 2012–2017 that were previously uploaded to the GenBank database by our laboratory (*Lyu et al., 2019*). The whole genomes of these viruses were analyzed along with all complete H3N2 CIV genomes publicly available in GenBank and the GISAID database (https://www.gisaid.org/) (n=229), and we constructed the maximum-likelihood phylogenic trees of eight viral gene segments (*Figure 1—figure supplements*

*3–10*). The inferred trees for all genomic segments exhibited a similar topology; thus, we grouped the viruses into six clades (clades 0–5) according to HA phylogeny, with several viruses consistently clustering together with high posterior probability values (bootstrap values ≥70). Clade 0 contains viruses isolated in China from 2006 to 2007, and clade 1 contains isolates exclusively from South Korea from 2007 and 2012. Clade 2 represents viruses isolated in China from 2009 to 2016, Thailand in 2012, and the United States in 2017; and clade 3 contains viruses collected in South Korea from 2012 and 2013. Clade 4 contains strains isolated in the United States from 2015 to 2017 and South Korea in 2015, and clade 5 encompasses isolates from China from 2016 to 2019 and the United States from 2017 to 2018. Most H3N2 CIVs after 2019 isolated in China have formed a further subclade 5.1 (*Figure 1A*). Comparing sequences of H3N2 CIVs with human influenza viruses and ancestral avian influenza viruses showed that, compared with ancestral avian influenza viruses, H3N2 CIVs that were initially introduced to dogs possessed several substitutions identical to human influenza viruses with high frequencies (>90%). A noteworthy observation was the number of human-like amino-acid substitutions that had gradually accumulated during the evolution of H3N2 CIVs in dogs and increased significantly after 2016 (*Figure 1B*). These results indicated that H3N2 CIVs may have increased their adaptability to humans during their evolution in dogs.

## Humans lack immunity to H3N2 CIVs

We performed an antigenicity test for representative H3N2 CIVs from different clades. *Supplementary file 1* and *Figure 1C* show that H3N2 CIVs continuously occurred antigenic changes worldwide. The cross-reactive titers between different antigenic groups were greater than or equal to fourfold lower than those of homologous reactions. The reaction patterns of clade 0 and clade 1 were similar and belonged to antigenic group A. Some clade 2 viruses were in antigenic group B, while other clade 2 viruses belonged to antigenic group C. The antigenicity of clade 3 and clade 4, which belong to antigenic group D, is different from other clades. Clade 5 included viruses belonging to antigenic groups E, F, or G. The co-circulation of different antigenic group viruses in recent years increased the difficulty of preventing and controlling canine influenza viruses. Additionally, we found that no H3N2 CIV was recognized by antisera to H3N2 human seasonal influenza virus in the hemagglutinin inhibition (HI) and neuraminidase inhibition (NI) assays (*Supplementary file 1* and *Supplementary file 2*).

To further investigate whether humans have existing immune protection against H3N2 CIVs, we tested sera collected from children (≤10 year old, n=100), adults (25–53 year old, n=100), and elderly adults (≥60 year old, n=100) against four viruses (BJ/1230/16, human seasonal H3N2 influenza; Cn/BJ/38/16, group C; Cn/FJ/1109/18, group D and Cn/GZ/011/19, group E) for HI, NI, and microneutralization (MNT) antibodies, as described previously (*Potter and Oxford, 1979*; *Rowe et al., 1999*; *Sandbulte et al., 2009*). We found that 15.0%, 1.0%, 1.0%, and 2.0% of children; 8.0%, 0.0%, 1.0%; 1.0% of adults; and 5.0%, 0.0%, 0.0%, and 1.0% of elderly adults had HI antibody titers of ≥40 to BJ/1230/16, Cn/BJ/38/16, Cn/FJ/1109/18, and Cn/GZ/011/19, respectively (*Table 1*). In addition, 26.0%, 2.0%, 2.0%, and 3.0% of children had NI antibody titers of ≥10, and 12.0%, 1.0%, 2.0%, and 1.0% of adults and 12.0%, 1.0%, 1.0%, and 2.0% of elderly adults had NI antibody titers of ≥10 to BJ/1230/16, Cn/BJ/38/16, Cn/FJ/1109/18, and Cn/GZ/011/19, respectively. Furthermore, 14.0%, 1.0%, 1.0%, and 2.0% of children had MNT antibody titers of ≥40; 6.0%, 0%, 0%, and 0% of adults; and 4.0%, 0%, 0%, and 0% of elderly adults had MNT antibody titers of ≥80 to BJ/1230/16, Cn/BJ/38/16, Cn/FJ/1109/18, and Cn/GZ/011/19, respectively.

These results indicated that human populations lack immunity to H3N2 CIV, and even preexisting immunity derived from the present human seasonal influenza viruses cannot provide protection against H3N2 CIVs.

## H3N2 CIVs obtained human-type receptor-binding properties and their acid stability increased stepwise

Our genetic analysis found that humanized adaptive mutations increased significantly along with the prevalence of H3N2 CIVs in dogs, while humans lacked preexisting immunity to the H3N2 CIVs, indicating that H3N2 CIVs might spread in populations once they are adapted to humans. Therefore, we further evaluated the potential threat of H3N2 CIVs to public health. The binding preference of HA for the host Saα2 6 Gal receptor and low activation pH are critical determinants for cross-species transmission of influenza virus to humans (*Connor et al., 1994*; *Matrosovich et al., 2000*). Therefore, we

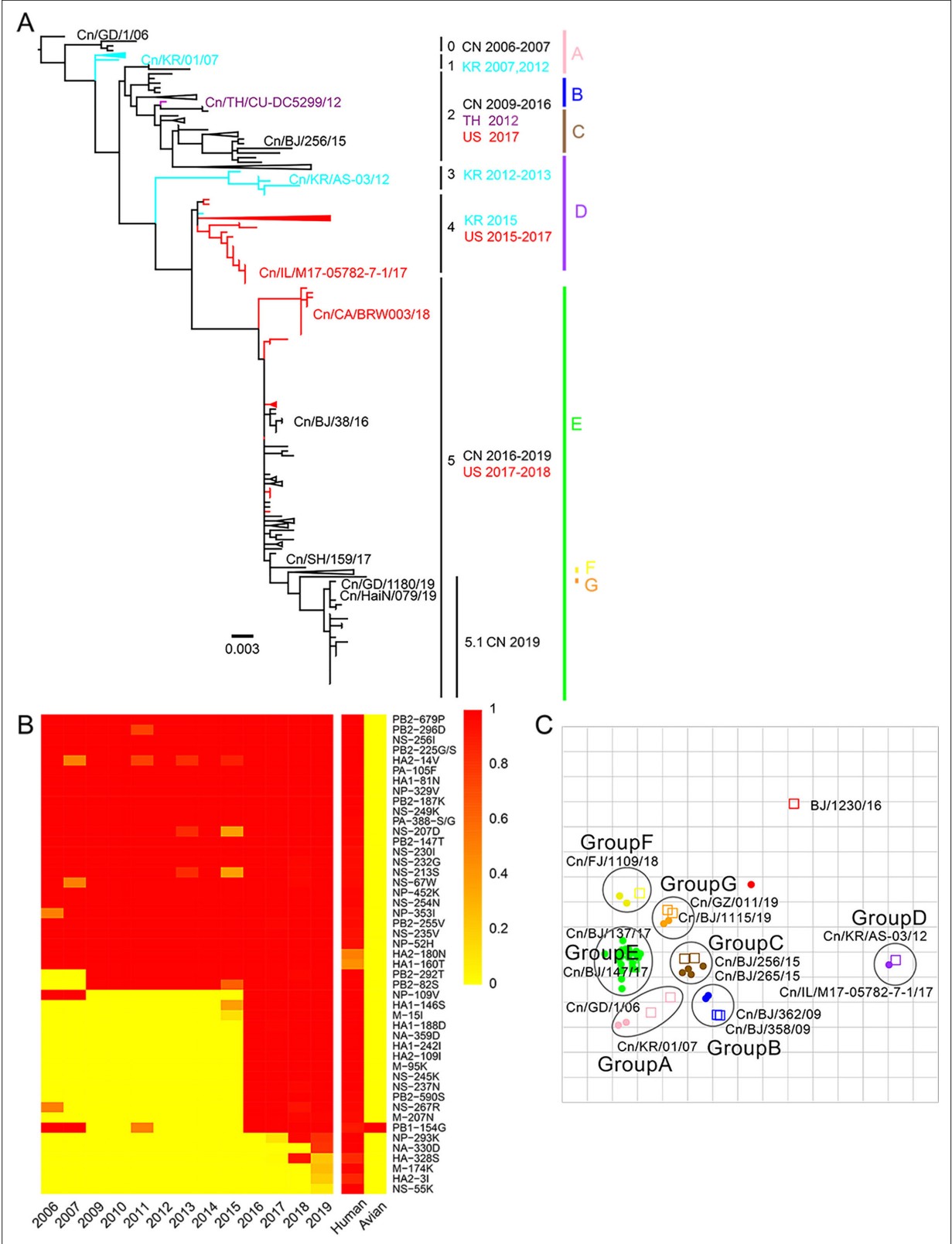

**Figure 1.** Genetic and antigenic characterization of H3N2 CIVs. (**A**) Maximum-likelihood phylogenetic tree of hemagglutinin (HA) genomic segment of H3N2 CIVs. The phylogenetic tree of the HA gene was estimated using genetic distances calculated by maximum likelihood under the GTRGAMMA +I model. Viruses with full names in the tree in (**A**) were selected for animal experiments. Black, red, dark purple, and aqua blue indicate H3N2 CIVs from China, the United States, Thailand, and South Korea, respectively. A, B, C, D, E, F, and G represent different antigen groups of H3N2 CIVs, respectively.

*Figure 1 continued on next page*

*Figure 1 continued*

A full detailed HA gene tree with a consistent topology is shown in *Figure 1—figure supplement 3* (scale bar is in units of nucleotide substitutions per site). (**B**) Prevalence of mammalian adaption markers among H3N2 CIVs. The sequences of H3N2 CIVs available in NCBI have compared with avian and human influenza A viruses. Color indicates the frequency of indicated substitutions in H3N2 CIVs for each indicated time period. (**C**) Antigenic map based on the HI assay data. Open squares and filled circles represent the positions of antisera and viruses, respectively. A k-means clustering algorithm identified clusters. Strains belonging to the same antigenic cluster are encircled with an oval. The vertical and horizontal axes both represent antigenic distance. The spacing between grid lines is 1 unit of antigenic distance, corresponding to a twofold dilution of antiserum in the HI assay. Details of the hemagglutinin inhibition (HI) assay data are shown in *Supplementary file 1*.

The online version of this article includes the following figure supplement(s) for figure 1:

**Figure supplement 1.** Map showing pet hospital and kennel density in China and geographic location of influenza surveillance in dogs from 2012–2019.

**Figure supplement 2.** Isolation rate (%) of H3N2 CIVs during 2012–2019 from dogs with respiratory symptoms.

**Figure supplement 3.** Hemagglutinin (HA) phylogenetic tree of fully sequenced H3N2 CIVs worldwide from 2006 to 2019.

**Figure supplement 4.** NA phylogenetic tree of fully sequenced H3N2 CIVs worldwide from 2006 to 2019.

**Figure supplement 5.** PB2 phylogenetic tree of fully sequenced H3N2 CIVs worldwide from 2006 to 2019.

**Figure supplement 6.** PB1 phylogenetic tree of fully sequenced H3N2 CIVs worldwide from 2006 to 2019.

**Figure supplement 7.** PA phylogenetic tree of fully sequenced H3N2 CIVs worldwide from 2006 to 2019.

**Figure supplement 8.** NP phylogenetic tree of fully sequenced H3N2 CIVs worldwide from 2006 to 2019.

**Figure supplement 9.** M phylogenetic tree of fully sequenced H3N2 CIVs worldwide from 2006 to 2019.

**Figure supplement 10.** NS phylogenetic tree of fully sequenced H3N2 CIVs worldwide from 2006 to 2019.

examined the receptor-binding preference of 11 H3N2 canine influenza viruses isolated from 2006–2019 (*Figure 2—figure supplement 1*). We noticed that, compared with the H3N2 avian influenza virus Dk/KR/JS53/04 and H3N2 CIVs from clades 0, 1, 2, and 3, which only recognized α–2,3-linked sialosides (*Figure 2A*), H3N2 CIVs belonging to clades 4, 5, and 5.1, represented by Cn/US/M17/17, Cn/SH/159/17, and Cn/HaiN/079/19, showed dual binding specificity to both α–2,3- and α–2,6-linked sialosides. The HA acid stability of clade 5 viruses (activation pH 5.3) was higher than that of clade 0, 1, 2, 3, and 4 viruses (activation pH 5.4) (*Figure 2B* and *Figure 2—figure supplement 2*). Compared with the H3N2 CIVs that were circulating before 2019 (activation pH 5.3), the viruses belonging to

**Table 1.** Cross-reactive antibody responses against influenza A (H3N2) viruses of human sera collected in China.

| Age group (year) | Antigen | %HI titer ≥40 (95% CI*) | %NI titer ≥10 (95% CI*) | %NT titer ≥40 for children or ≥80 for adults (95% CI*) |
|---|---|---|---|---|
| | BJ/1230/16 (human) | 15.0 (7.9–22.1) | 26.0 (17.4–34.6) | 14.0 (7.2–20.8) |
| | Cn/BJ/38/16 | 1.0 (0–2.9) | 2.0 (0–4.7) | 1.0 (0–2.9) |
| | Cn/FJ/1109/18 | 1.0 (0–2.9) | 2.0 (0–4.7) | 1.0 (0–2.9) |
| ≤10, n=100 | Cn/GZ/011/19 | 2.0 (0–4.7) | 3.0 (0–6.3) | 2.0 (0–4.7) |
| | BJ/1230/16 (human) | 8.0 (2.7–13.3) | 12.0 (5.6–18.4) | 6.0 (1.4–10.6) |
| | Cn/BJ/38/16 | 0 | 1.0 (0–2.9) | 0 |
| | Cn/FJ/1109/18 | 1.0 (0–2.9) | 2.0 (0–4.7) | 0 |
| 25–53, n=100 | Cn/GZ/011/19 | 1.0 (0–2.9) | 1.0 (0–2.9) | 0 |
| | BJ/1230/16 (human) | 5.0 (0.7–9.3) | 12.0 (5.6–18.4) | 4.0 (0.2–7.8) |
| | Cn/BJ/38/16 | 0 | 1.0 (0–2.9) | 0 |
| | Cn/FJ/1109/18 | 1.0 (0–2.9) | 1.0 (0–2.9) | 0 |
| ≥60, n=100 | Cn/GZ/011/19 | 1.0 (0–2.9) | 2.0 (0–4.7) | 0 |

*Confidence interval.

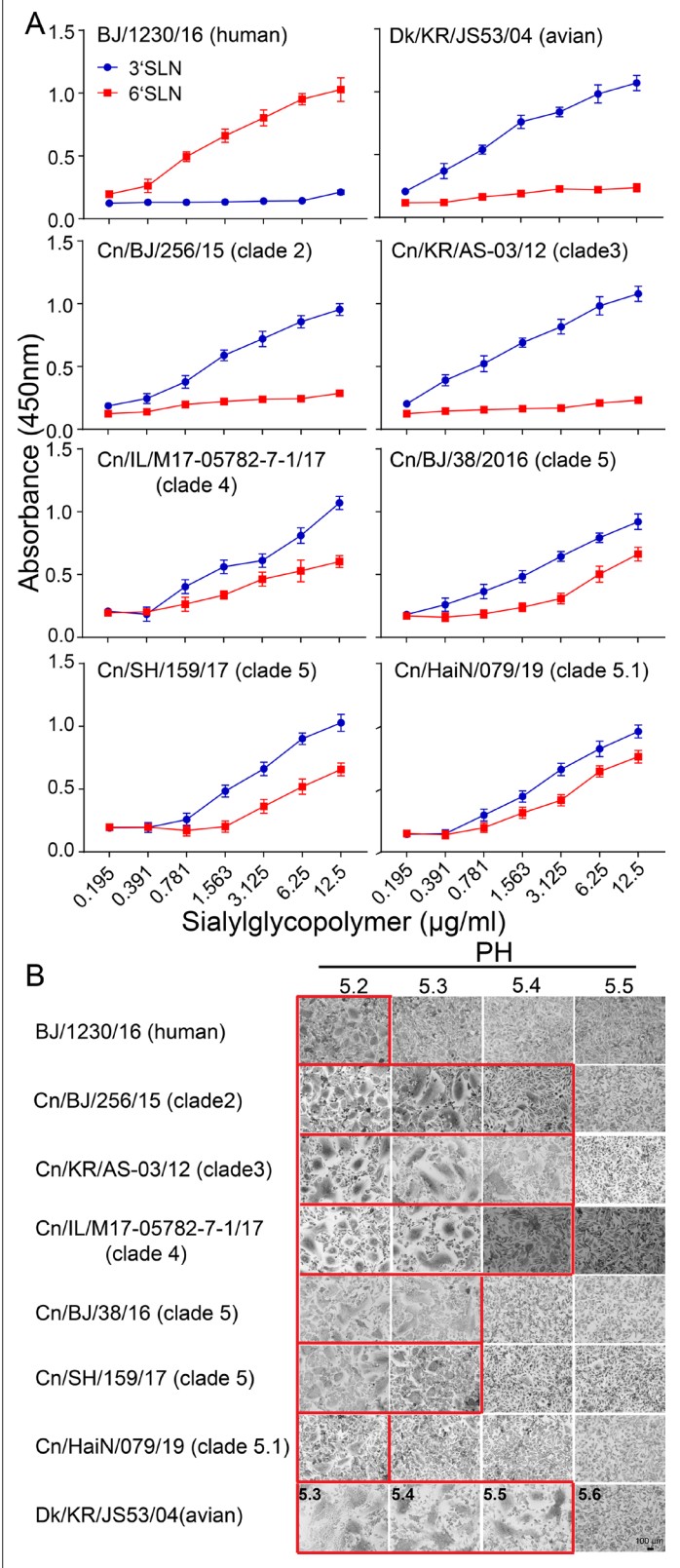

**Figure 2.** Binding specificities toward α–2, 3-, or α–2, 6-linked sialic acid receptors and hemagglutinin (HA) acid stability. (**A**) Characterization of receptor-binding properties of H3N2 CIVs. Direct binding of the virus to sialylglycopolymers containing either 2,3-linked (blue) or 2,6-linked (red) sialic acids was tested (n=3 biological replicates and n=3 technical replicates). Values are expressed as means ± standard deviations (SD). (**B**) HA

*Figure 2 continued on next page*

*Figure 2 continued*

activation pH measured by syncytia assay. Representative fields of cells infected with the indicated viruses and exposed to pH 5.2, 5.3, 5.4, 5.5, or 5.6 are shown. Scale bar, 100µm. The experiments were repeated three times, with similar results.

The online version of this article includes the following figure supplement(s) for figure 2:

**Figure supplement 1.** Characterization of the receptor-binding properties of H3N2 CIVs.

**Figure supplement 2.** Syncytia assay results for H3N2 CIVs.

clade 5.1 circulating after 2019 had higher HA acid stability (activation pH 5.2) and identical HA fusion pH to that of the human H3N2 virus.

The replication efficiency of CIVs exhibiting different receptor-binding specificity and HA acid stability were evaluated in vitro. In A549 cells, the titers of clade 5 and 5.1 viruses were significantly

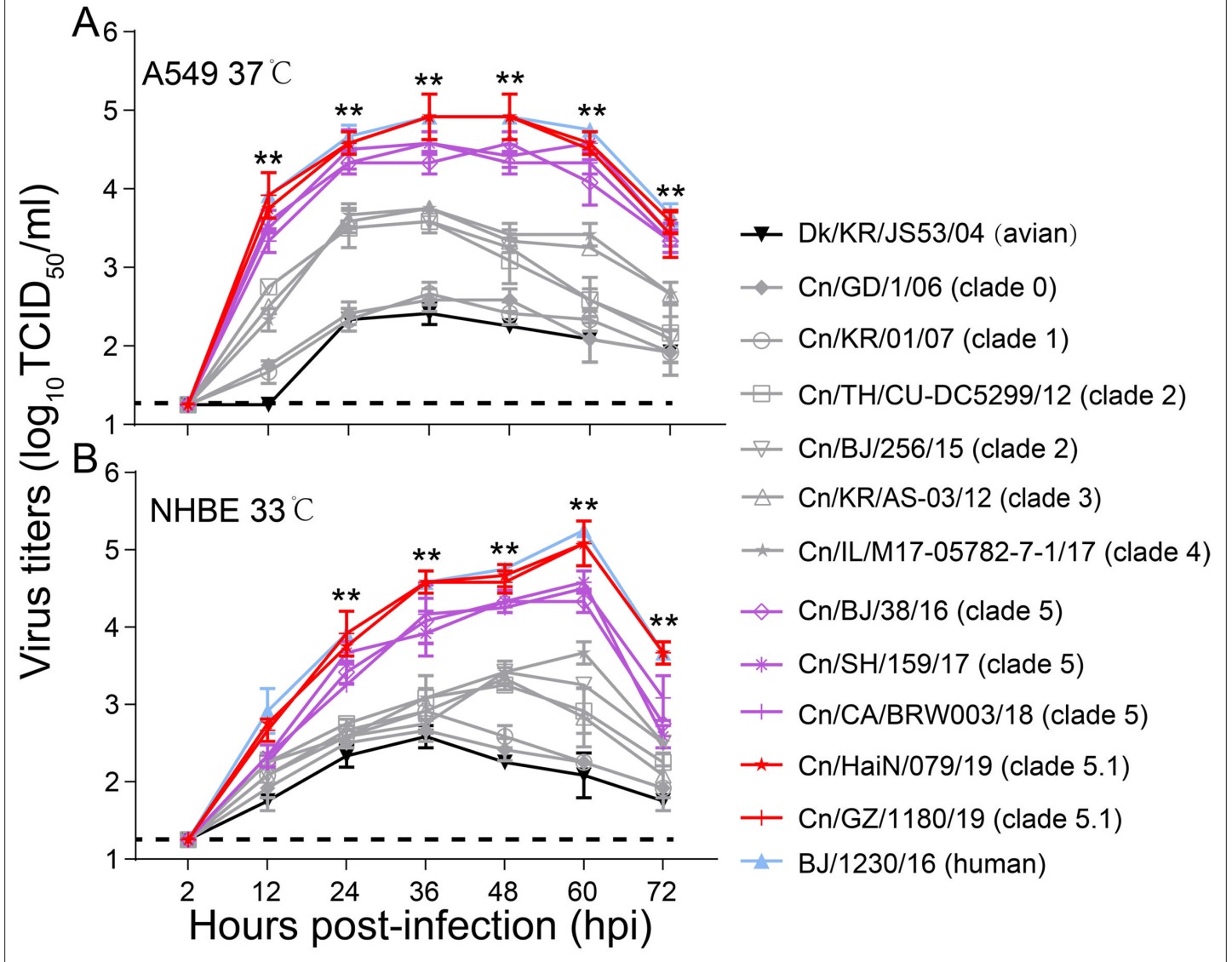

**Figure 3.** Viral growth properties in A549 (**A**) and NHBE (**B**) cells. Cells were infected with indicated viruses at MOI of 0.01 and incubated at 37 °C or 33 °C. Supernatants were harvested at the indicated time points, and the virus titers were determined in MDCK cells. Values are expressed as means ± standard deviations (SD) (n=3 biological replicates and n=3 technical replicates). **p<0.01,statistical significance was assessed using two-way ANOVA, the titers of BJ/1230/16 (human seasonal H3N2 virus) and clade 5 or 5.1 viruses were significantly higher than other viruses. The dashed black lines indicate the lower limit of detection.

higher (up to nearly 100-fold higher) than those of other clades over 12 to 72 hr (p<0.01) (*Figure 3A*), and clade 5.1 viruses showed comparable virus outputs with the human seasonal H3N2 virus at each time point. Infection of NHBE cells with H3N2 CIVs produced similar progeny results. The titers of clade 5 and 5.1 viruses in NHBE cells were significantly (up to nearly 100-fold higher) higher than those of viruses from other clades between 24 hpi and 72 hpi (p<0.01), and clade 5.1 viruses and human seasonal H3N2 virus had also replicated to similar levels at each time point (*Figure 3B*). Collectively, H3N2 CIVs obtained human-type receptor binding properties, and their HA acid stability and replication ability in human cells increased stepwise during their circulation in the dog population.

## Improved replication and transmissibility of H3N2 CIVs in dogs

To evaluate the infectivity and transmission ability of H3N2 CIVs in dogs, we inoculated intranasally six dogs with $10^6$ TCID$_{50}$ of each virus strain (*Figure 4—figure supplement 1*). Twenty-four hours later, three dogs inoculated with each virus strain were individually paired and cohoused with a direct-contact dog. At 4 dpi, nasal turbinates, tracheas, lungs, and tonsils were collected from another three inoculated dogs in each infection group for virus titration. We found that the H3N2 avian influenza virus could not be detected in dogs at 4 dpi, while H3N2 CIVs from clades 5 and 5.1 replicated efficiently in both the upper (nasal turbinate and trachea) and lower (lung) respiratory tracts and tonsils of dogs, where they were present in significantly higher amounts than the other clade viruses (p<0.01) (*Figure 4A*). Additionally, we monitored the clinical signs of the three inoculated dogs used for the transmission experiment for 14 days, and we found that H3N2 avian influenza virus (Dk/KR/JS53/04) and H3N2 CIVs (from clades 0, 1, 2, 3, and 4) caused only mild clinical signs with mean clinical scores ranging from 0.5 to 2.5 and a mean body temperature ranging from 37.7 to 39.7°C (*Figure 4—figure supplement 1* and *Figure 4B and C*). However, infection with H3N2 CIVs from clades 5 and 5.1 resulted in more severe clinical symptoms such as pyrexia, sneezing, wheezing, and coughing, with a higher mean clinical score ranging from 2.8 to 3.3 and a higher mean body temperature ranging from 39.9 to 40.2°C. Furthermore, the virus identification of nasal swabs showed that H3N2 CIVs were efficiently transmitted to all naïve dogs by direct contact (*Figure 4D* and *Figure 4—figure supplement 1*). In contrast, the H3N2 avian influenza virus was not transmitted between dogs. H3N2 CIVs (from clade 0, 1, 2, 3, and 4) were detected in two of the three naïve animals and all contact animals at 4 dpi and 6 dpi, respectively, and seroconversion was detected in 2/3 contacts (1:80 to 1:160) or all contacts (1:160 to 1:320) (*Supplementary file 3*). Noteworthy, clade 5 and 5.1 viruses, represented by Cn/BJ/38/16, Cn/SH/159/17, Cn/CA/BRW003/18, Cn/HaiN/079/19, and Cn/GZ/1180/19 were transmitted to all three contact animals at 2 dpi with seroconversion in all contacts (1:320 to 1:640), which was earlier than clade 0, 1, 2, 3, and 4 viral transmissions to naïve animals. Furthermore, clade 5 and 5.1 viruses also replicated more efficiently in all donors than other clade viruses (p<0.01). Thus, the replication and transmissibility of H3N2 CIVs gradually increased in dogs.

## H3N2 CIVs acquired efficient aerosol transmissibility in a ferret model after the 2016

To further evaluate the potential risk of H3N2 CIVs to public health, we examined the replication and transmission of H3N2 viruses in a ferret model. A group of six ferrets was inoculated intranasally with $10^6$ TCID$_{50}$ of each virus strain. Twenty-four hours later, three inoculated ferrets for each virus strain were individually paired. An uninfected animal was housed in a wire-frame cage adjacent to the infected ferret to assess aerosol spread. Virus detection in nasal washes of the aerosol-spread animal showed that the H3N2 avian influenza virus (Dk/KR/JS53/04) and H3N2 CIVs (from clades 0, 1, 2, 3, and 4) did not transmit to ferrets through respiratory droplets (*Figure 5*). Clade 5 viruses, represented by Cn/BJ/38/16, Cn/SH/159/17, and Cn/CA/BRW003/18, transmitted to all naïve three ferrets via respiratory droplets by 6 dpi, and seroconversion was detected in all aerosols (1:160 to 1:320) (*Supplementary file 4*). More importantly, clade 5.1 viruses, represented by Cn/HaiN/079/19 and Cn/GZ/1180/19, were able to transmit to all three naïve ferrets as early as 4 dpi, with seroconversion in 3/3 aerosols (1:320 to 1:640). In addition, clade 5 and 5.1 viruses also replicated more efficiently than other clade viruses in all donors. Clade 5.1 viruses showed comparable virus outputs and transmissibility with the human seasonal H3N2 virus in the ferret model. Collectively, the evidence shows H3N2 CIVs obtained aerosol transmissibility during their evolution in dogs.

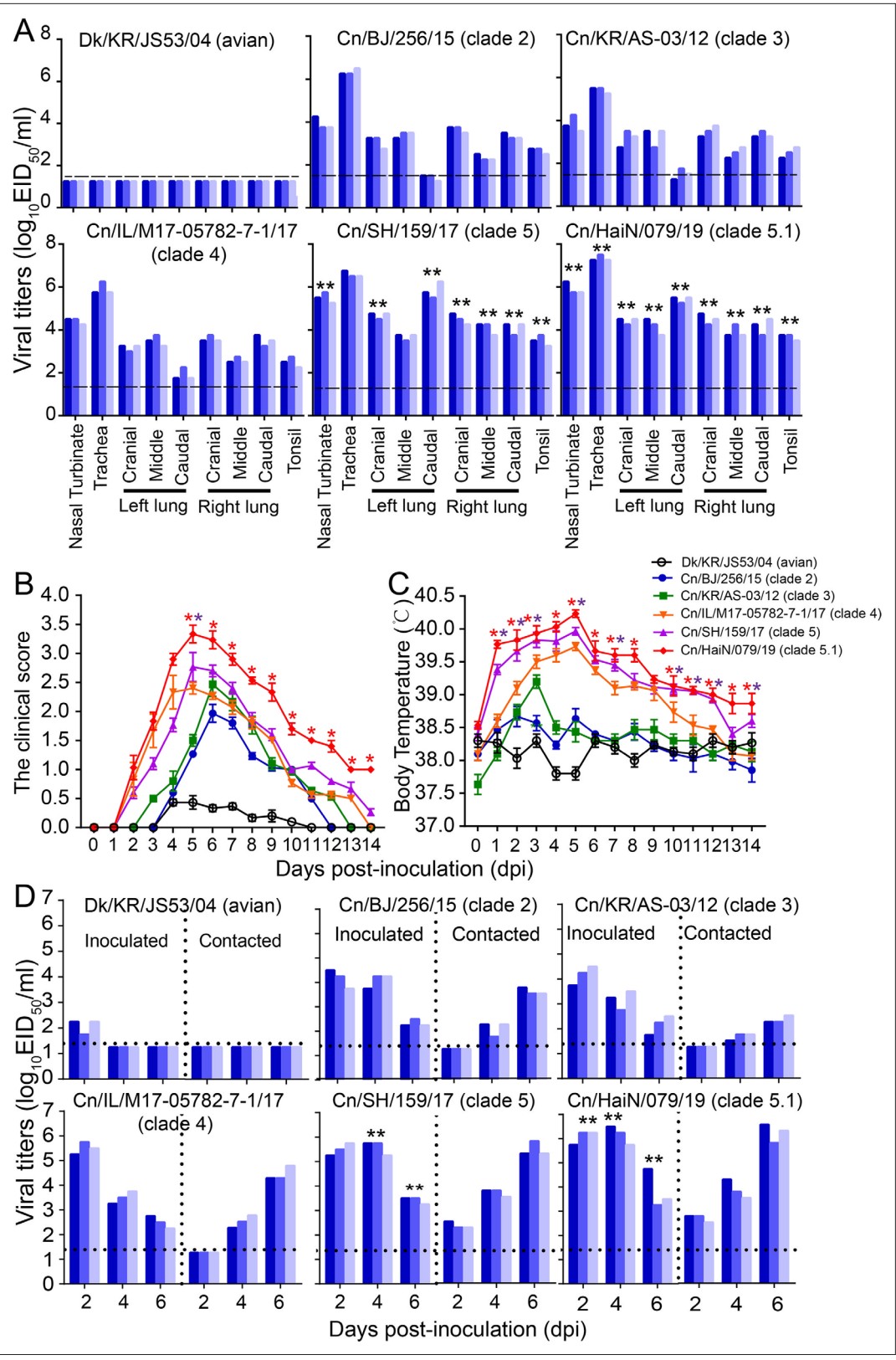

**Figure 4.** Infectivity and transmissibility of H3N2 CIVs in dogs. (**A**) Virus replication in the indicated organ. Three dogs were infected intranasally with $10^6$ $EID_{50}$ of each virus and euthanized at 4 dpi for virus titration. Each color bar represents the virus titer of an individual animal. (**B**) Clinical symptoms score of dogs infected with H3N2 CIVs. (**C**) Body temperatures of dogs infected with H3N2 CIVs. The results are shown as the means ± standard deviations

*Figure 4 continued on next page*

*Figure 4 continued*

(n = 3) (*p<0.05; **p<0.01). (**D**) Direct contact transmission of H3N2 CIVs in dogs. Each virus was tested with a total of three donors that were in direct contact with each group. Dogs housed in the same cage are denoted by the same color. The dashed black lines indicate the lower limit of detection. Statistical significance of clade 5 or clade 5.1 viruses relative to other viruses in the inoculated animals was assessed using two-way ANOVA (*p<0.05; **p<0.01).

The online version of this article includes the following figure supplement(s) for figure 4:

**Figure supplement 1.** Direct contact transmission of H3N2 CIVs in dogs.

## Molecular determinants associated with efficient transmission reside in the HA and PB1 genes

Since recent strains have increased replication and transmissibility in dogs and have gradually acquired 100% aerosol transmission ability in the ferret model, we further determined the molecular mechanisms responsible for the enhanced replication and transmission of H3N2 CIVs in mammals. We used Cn/BJ/256/15 (clade 2) as the backbone to generate reassortant viruses by individually replacing all genes from the Cn/HaiN/079/19 (clade 5.1) virus and testing the viral replication and transmissibility among dogs and ferrets. We found that PB1 and HA genes significantly enhanced the replication and transmission of single-gene reassortant viruses in both dogs and ferrets (*Supplementary file 5* and *Supplementary file 6*), and seroconversion was detected in all contact dogs (1:160 to 1:320) and 2/3 aerosol ferrets (1:80 to 1:160). Next, we identified 15 conserved amino acid variations in the HA protein and PB1 protein of H3N2 CIVs that emerged in 2016–2019 (*Figure 6—figure supplement 1*). Among them, HA-146S, 188D, and PB1-154G were also highly enriched (>90%) among human H3N2 influenza A viruses isolated from 2006–2019 (*Figure 6A*). In addition, HA-16S was detected at a high frequency (>81%) among the H3N2 CIVs isolated after 2019, while all H3N2 CIVs before 2018 possessed HA-G16 (*Figure 6A*). Next, we evaluated the effect of these substitutions in receptor binding property assays and acid stability and thermal stability experiments. We found that the introduction of the HA-G146S enhanced binding to α–2,6-linked sialosides in Cn/BJ/256/15 (*Figure 6B*). The introduction of HA-G16S and HA-N188D increased the HA acid and temperature stability of Cn/BJ/256/15 (*Figure 6C and D*). Introduction of PB1-D154G increased the polymerase activity of Cn/BJ/256/15 (*Figure 6E*).

We then focused on these four mutations and evaluated the replication and respiratory droplet transmission ability of rgHA-(G16S), rgHA(G146S), rgHA(N188D), rgPB1(D154G), and rgHA(G16S, G146S, N188D)PB1(D154G) viruses in ferrets. We found that rgHA(G146S), rgHA(N188D), and rgPB1(D154G) viruses were transmitted to two of the three ferrets with seroconversion in 2/3 aerosols (1:160 to 1:320), and rgHA-(G16S), and rgHA(G16S, G146S, N188D)PB1(D154G) viruses were transmitted to all three ferrets with seroconversion in 3/3 aerosols (1:320 to 1:640) (*Figure 6F*). In addition, the viral titers in the nasal washes from the rgHA(G16S, G146S, N188D) PB1(D154G)-inoculated animals were higher than those of the other single substitution viruses (p<0.05) and were similar to that of the transmissible wild-type virus (Cn/HaiN/079/19). These results indicated that HA-G16S, G146S, N188D, and PB1-D154G are crucial for the replication and efficient transmissibility of H3N2 CIVs in ferrets.

## Discussion

By evaluating the biological characteristics of avian-origin H3N2 CIVs isolated in dogs in different years, we found that effective infection and transmission in dogs were not intrinsic properties of H3N2 CIVs but appeared through stepwise adaptation. Of note, threats to human health might increase during H3N2 CIVs' adaptation to dogs. Specifically, we observed changes in receptor binding specificity, from viruses recognizing only α–2,3-linked sialosides to those recognizing both α–2,3- and α–2,6-linked sialosides, as well as gradually increased HA acid stability and replication in human airway epithelial cells and ferrets and the ability to be transmitted by aerosol among ferrets. In addition, humans lack immunity to the H3N2 CIVs, and increased isolation of H3N2 CIVs in the dog population might increase the chance of transmission to humans. Therefore, we suggested that dogs are potential intermediate hosts in which avian influenza viruses can adapt to humans.

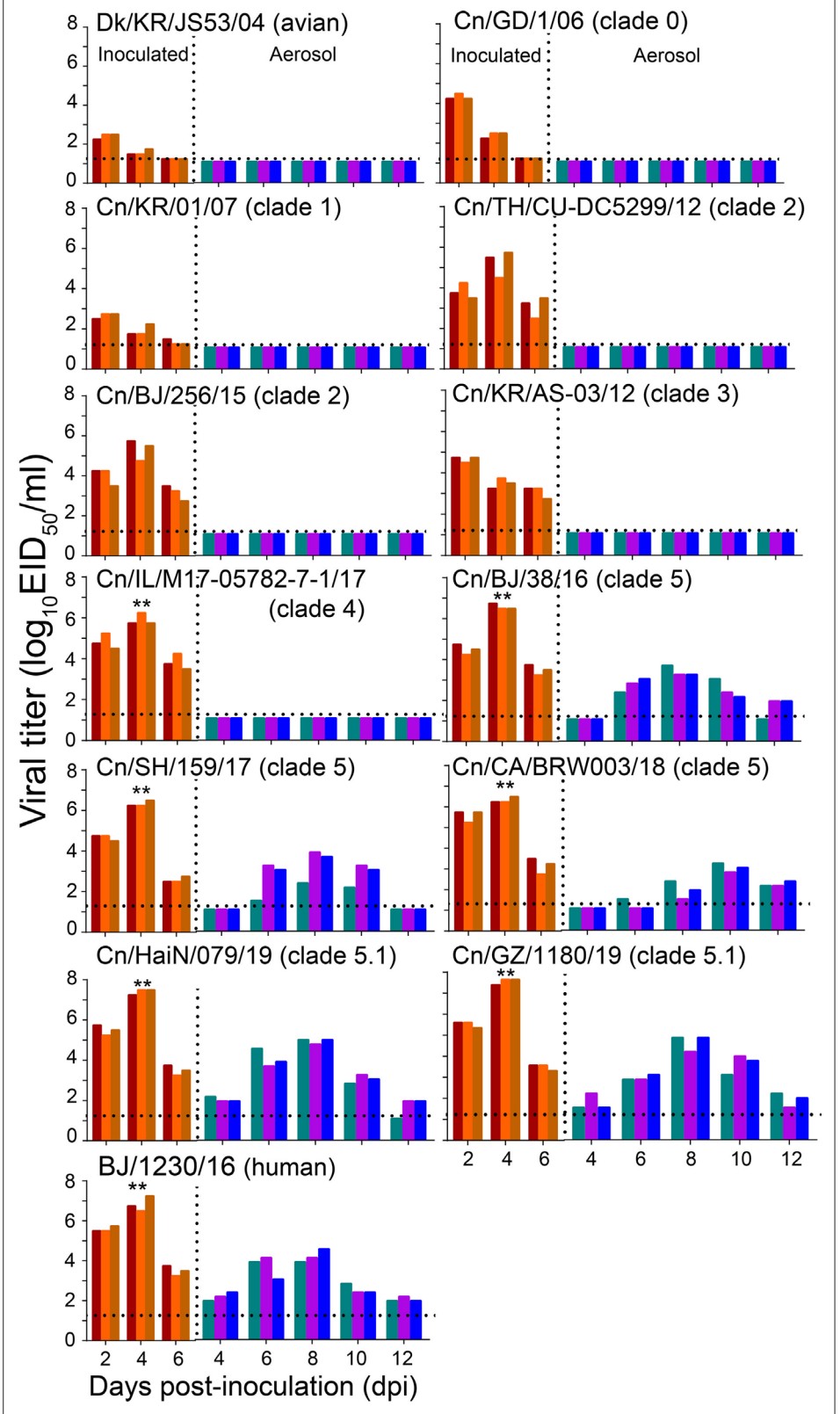

**Figure 5.** Respiratory droplet transmission of H3N2 CIVs in ferrets. Groups of three ferrets were infected intranasally with $10^6$ $EID_{50}$ of indicated viruses and then housed separately in solid stainless-steel cages within an isolator. The next day, three uninfected respiratory droplet contact animals were individually housed in a wire-frame cage adjacent to the infected ferret. Nasal washes were collected every other day from all animals for

*Figure 5 continued on next page*

*Figure 5 continued*

virus-shedding detection from day 2 of the initial infection. Each color bar represents the virus titer of an individual animal. No data are displayed when no virus was detected in any of the groups. Dashed lines indicate the lower limit of virus detection. Statistical significance of the human influenza virus (BJ/1230/16) and clade 5.1 or clade 5 viruses relative to other viruses in the inoculated animals was assessed using two-way ANOVA (**p<0.01).

We found that H3N2 CIVs formed different clades after entering the dog population, and these clades were related to the time of virus isolation. Early H3N2 CIVs (clades 0 and 1) are not transmitted efficiently among dogs. After a particular opportunity, the avian H3N2 subtype influenza virus enters and gradually adapts to dogs. The H3N2 CIVs isolated after 2012 can reach 100% transmission in dogs. Clade 5 viruses isolated after 2016 were transmitted to all three contact animals within 2 dpi; thus the transmission time interval had shortened. Moreover, the replication ability of clade 5.1 strains generated after 2019 has further improved. Additionally, H3N2 CIV antigenicity has continuously evolved to form groups A, B, C, D, E, F, and G. The gradual improvement in viral infection and transmission ability and the continuous evolution of antigenicity promoted the prevalence of H3N2 CIV in dogs.

The high isolation rate of H3N2 CIVs in companion animal dogs might increase the opportunity for viral cross-species transmission to humans. We found that the number and frequency of substitutions identical to human influenza viruses increased in H3N2 CIVs during the adaptation process in dogs, and the number of homology sites increased significantly after 2016. Therefore, we evaluated the potential threat of H3N2 CIVs to public health and found that mutation of HA-G146S around the receptor-binding domain after 2016 caused a shift from the recognition of only α–2,3-linked sialosides (clade 0, 1, 2, and 3 viruses) to both α–2,3- and α–2,6-linked sialosides (clade 4, 5, and 5.1). Furthermore, the HA-G16S and HA-N188D mutations resulted in enhanced acid and thermostability, and the PB1-D154G mutation resulted in increased polymerase activity, which together resulted in the acquisition of aerosol-transmitting properties in ferrets. Notably, the proportions of HA-G146S, HA-N188D, and PB1-D154G are all higher than 90% in human influenza viruses. Therefore, during the gradual adaptation of CIVs in dogs, clade 5 CIVs with high adaptability to mammals were selected. Additionally, we found that clade 5 CIVs further adapted after 2019. The HA-G16S mutation, which improves acid stability, further improved the replication ability and shortened the transmission time of CIVs in ferrets.

Although our results suggest that, after the avian-origin H3N2 influenza viruses were introduced to dogs, they adapted to canines and might represent an increased threat to public health, no studies have yet reported human infection with H3N2 CIV. Additionally, although influenza viruses that can be transmitted between humans possess transmissibility via respiratory droplets to ferrets, the reverse scenario does not necessarily occur. Nevertheless, to lower the prevalence of novel influenza viruses that are a potential public health threat, controlling the prevailing H3N2 CIVs in dogs and continuous monitoring of their biological characteristics should be implemented.

# Materials and methods
## Viruses and cells

Nasopharyngeal swabs were collected from dogs with signs of respiratory disease in animal hospitals or kennels in nine provinces or municipalities of China (Liaoning, Beijing, Tianjin, Jiangsu, Shanghai, Shanxi, Guangdong, Fujian, and Hainan) between 2017 and 2019. Nasopharyngeal swabs were placed in 1.0 ml of transmission medium [50% (vol/vol) glycerol in PBS] containing antibiotics, as previously described (*Zhang et al., 2008*). We amplified the matrix gene by real-time reverse transcription (RT) PCR using the Influenza A Virus V8 Rapid Real-Time RT-PCR Detection Kit (Beijing Anheal Laboratories Co. Ltd., http://anheal.company.weiku.com) and isolated and identified virus isolates using methods described previously (*Sun et al., 2013*). The Chinese H3N2 CIVs that we isolated before 2017 and human seasonal H3N2 virus A/Beijing/1230/2016 (BJ/1230/16) were described previously (*Lyu et al., 2019*; *Sun et al., 2013*; *Sun et al., 2021*).

HEK293T cells, human lung carcinoma cells (A549), Vero cells, and Madin-Darby canine kidney (MDCK) cells were obtained from the American Type Culture Collection (ATCC) and cultured in Dulbecco's modified Eagle's medium (DMEM Gibco) supplemented with 10% fetal bovine serum

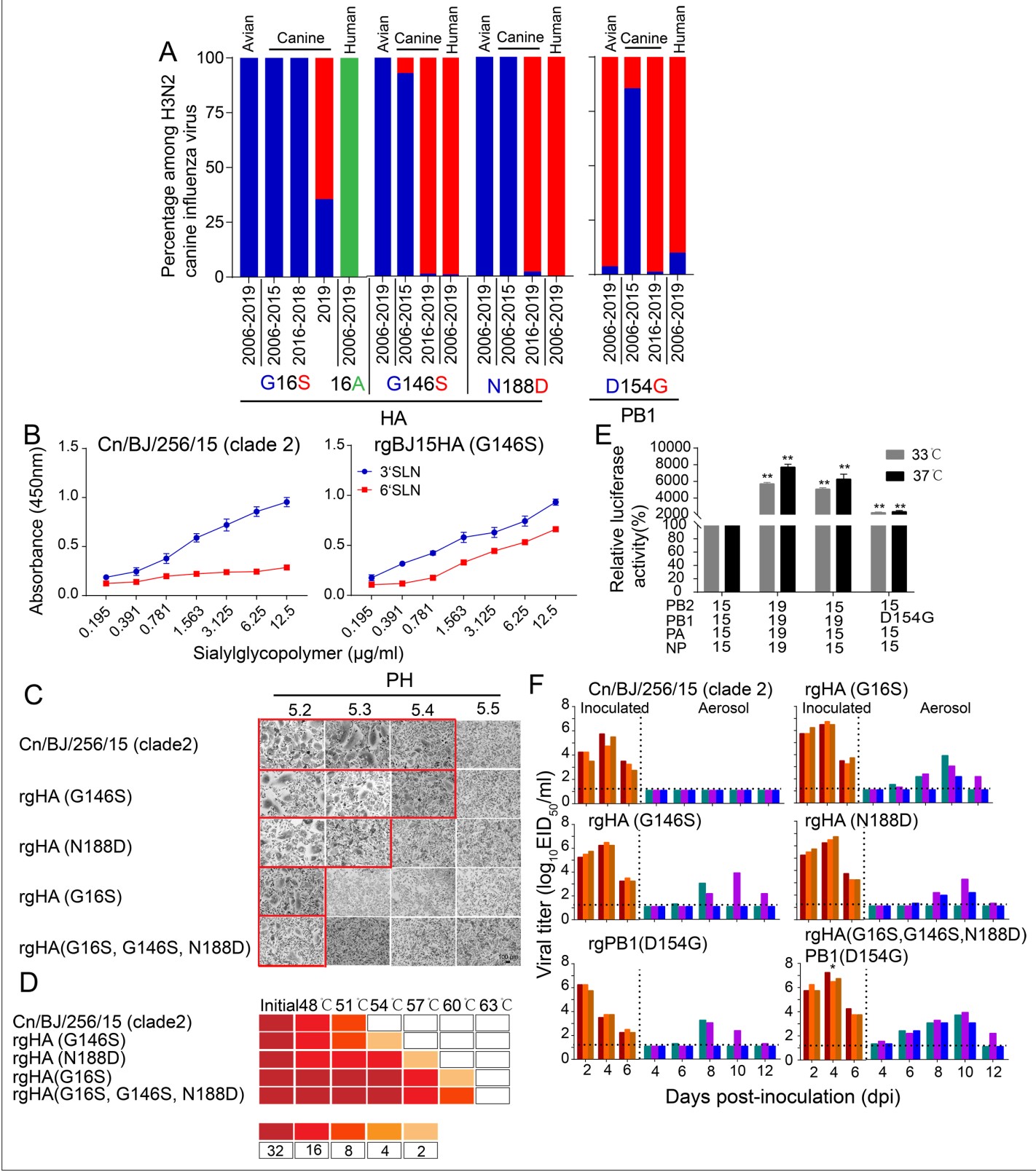

**Figure 6.** HA-G16S, G146S, N188D, and PB1-D154G mutations in H3N2 CIVs were the minimal molecular change required to facilitate the efficient aerosol transmissibility of the non-transmissible clade 2 virus. (**A**) Detection frequency of G/S at hemagglutinin (HA) residue 16 and 146, N/D at HA1 residue 188 (n=437), and D/G at PB1 residue 154 (n=437) in avian, canine, and human influenza viruses. Amino acid residues are colored blue, red, and green, respectively. (**B**) Identification of mutations that confer binding to human-type receptors (n=3 biological replicates and n=3 technical replicates).

*Figure 6 continued on next page*

*Figure 6 continued*

Values are expressed as means ± standard deviations (SD). The wildtype data (Cn/BJ/256/15) were reproduced from *Figure 2*. (**C**) Representative fields of Vero cells expressing the indicated HAs and exposed to pH 5.2, 5.3, 5.4, or 5.5 are shown. Scale bar, 100µm. The experiments were repeated three times, with similar results. (**D**) HA protein stability as measured by the ability of viruses to agglutinate CRBCs after incubation at indicated temperatures for 40 min. Colors indicate the hemagglutination titers upon treatment at various temperatures for 40 min, as shown in the legend. The experiments were repeated three times with similar results. (**E**) The effects of PB1 from 19(Cn/HaiN/079/19) and amino-acid substitutions at PB1 residue 154 on the viral polymerase activity were determined using a minigenome assay in 293T cells. Values shown are the mean ± SD of the three independent experiments and are standardized to those of 15 (Cn/BJ/256/15) measured at 37℃ (100%) and 33℃ (100%). Statistical significance was assessed using two-way ANOVA (**$p<0.01$). (**F**) Aerosol transmissibility of the mutation viruses in ferrets. The wildtype data (Cn/BJ/256/15) were reproduced from *Figure 5*. Statistical significance of rgHA(G16S, G146S, N188D)PB1(D154G) viruses relative to other single substitution viruses were assessed using two-way ANOVA (*$p<0.05$).

The online version of this article includes the following figure supplement(s) for figure 6:

**Figure supplement 1.** Detection frequency of the 15 amino acids that differed between 2006–2015 H3N2 CIVs and 2016–2019 H3N2 CIVs among HA (n=298) and PB1(n=298).

(FBS; Gibco), 100 U/ml of penicillin, and 100 µg/ml of streptomycin. NHBE cells (Lonza, Allendale, NJ, USA) were cultured in bronchial epithelial cell growth medium (Lonza) at the air-liquid interface, as previously described (*Hudy et al., 2010*; *Matrosovich et al., 2004*). All cells were maintained in a humidified incubator containing 5% $CO_2$ at 37 °C (HEK293T, A549, Vero cells, and MDCK cells) or 33 °C (NHBE cells). In addition, all of the cells used in the study tested negative in routine tests for Mycoplasma species using real-time PCR.

## Genetic and phylogenetic analyses

Viral gene amplification and sequencing were carried out as previously described (*Sun et al., 2010*). All previously published gene sequences from the H3N2 canine influenza virus were collected from the NCBI and the Global Initiative on Sharing Avian Influenza Data (https://www.gisaid.org). Phylogenetic analyses were performed on regions of the alignments containing the fewest gaps across sequences. These regions consisted of the following intervals: PB2, 82–2,061; PB1, 83–2,167; PA, 82–2,068; HA, 34–1,656; NP, 13–1,446; NA, 36–1,377; MP, 67–924; and NS, 28–838. RAxML was used to construct maximum likelihood phylogenies for each segment (*Stamatakis, 2014*), via CIPRES Science Gateway (*Miller et al., 2010*); 1,000 bootstrap replicates were run, and GTRGAMMA +I was used as the nucleotide substitution model.

## Generation of recombinant viruses by reverse genetics

The genome sequences of avian influenza virus A/duck/Korea/JS53/2004 (Dk/KR/JS53/04) and H3N2 CIVs A/canine/Guangdong/1/2006 (Cn/GD/1/06), A/canine/Korea/01/2007 (Cn/KR/01/07), A/canine/Thailand/CU-DC5299/2012 (Cn/TH/CU-DC5299/12), A/canine/Korea/AS-03/2012 (Cn/KR/AS-03/12), A/canine/Illinois/M17-05782-7-1/2017 (Cn/IL/M17-05782-7-1/17), and A/canine/California/BRW003/2018 (Cn/CA/BRW003/18) were downloaded from the NCBI influenza virus database and synthesized by Sangon Biotech Company (Shanghai, China). The eight gene segments of each virus were amplified by RT-PCR and cloned into the dual-promoter plasmid pHW2000, and then these viruses were generated using reverse genetics as previously described (*Sun et al., 2011*).

All eight gene segments from H3N2 CIVs A/canine/Beijing/0118-256/15(Cn/BJ/256/15, BJ15) and A/canine/Hainan/079/2019 (Cn/HaiN/079/19, HaiN19) were amplified by RT-PCR and then cloned into the dual-promoter plasmid pHW2000. Reverse-genetics systems for BJ15 and HaiN19 were then established, and single-gene reassortant viruses were rescued by reverse genetics. Mutations were introduced into the PB1 and HA genes using a QuikChange site-directed mutagenesis kit (Agilent) in accordance with the manufacturer's instructions. PCR primer sequences are available upon request. All constructs were sequenced to ensure the absence of unwanted mutations. The HA- and PB1-mutant viruses were generated using reverse genetics. Rescued viruses were detected using HA assays. Viruses were purified by sucrose density gradient centrifugation. Viral RNA was extracted and analyzed by RT-PCR, and each viral segment was sequenced.

## Antigenic analyses

Antigenic characteristics of H3N2 viruses were compared using an HI assay with ferret antisera raised against representative viruses. Ferret antisera raised against BJ/1230/16, Cn/GD/1/06 (group A),Cn/KR/01/07 (group A), Cn/BJ/358/09 (group B), Cn/BJ/362/09 (group B), Cn/BJ/256/15 (group C), Cn/BJ/265/15 (group C), Cn/KR/AS-03/12 (group D), Cn/IL/M17-05782-7-1/17(group D), Cn/BJ/137/17 (group E), Cn/BJ/147/17 (group E), Cn/FJ/1109/18 (group F), Cn/GZ/011/19 (group G), and Cn/BJ/1115/19 (group G) viruses were produced in our laboratory. All HI assays were performed following WHO guidelines and in duplicate. Antigenic properties were analyzed using the antigenic cartography methods described previously (*Smith et al., 2004*). The antigen cartography of the viruses was constructed using Antigenic Cartography software (http://www.antigenic-cartography.org/). Clusters were identified in the antigenic map by a k-means clustering algorithm using average weighting and k=3. We collected human sera from the First Medical Centre of Chinese PLA General Hospital in Beijing in 2021, and these sera were from children, adults, and elderly adults who did not have a fever. Informed consent to use the sera for influenza antibody detection was obtained. Sera were treated with *Vibrio cholerae* receptor-destroying enzyme (Denka-Seiken) before being tested for the presence of HI antibody with 1% chicken red blood cells (CRBC). The minimum cut-off value for the HI assay was 10. MNT assays were performed as reported previously (*Rowe et al., 1999*; *Hancock et al., 2009*). Titration of serum NI antibodies was performed by analyzing the neuraminidase (NA) activity of the human seasonal H3N2 virus (BJ/1230/16) and H3N2 canine influenza viruses (Cn/BJ/38/16; Cn/FJ/1109/18; Cn/GZ/011/19) in a 96-well plate format of the conventional thiobarbituric acid assay (*Sandbulte et al., 2009*). Briefly, twofold serial dilutions of sera were incubated overnight with the viruses mentioned above containing a fixed amount of NA activity in a solution of fetuin. Sialic acid released from fetuin was converted into a chromophore through a series of chemical reactions, and the amount was quantified after extraction into Warren off reagent (95% 1-butanol, 5% concentrated HCl) by measuring absorbance at 550 nm. The NI endpoint titer was defined as the reciprocal of the highest serum dilution that inhibited the NA signal by ≥50%.

## Receptor-binding assays

α–2,6 glycans (6′SLN: Neu5Acα2-6Galβ1-4GlcNAc β-SpNH-LC-LC-biotin) and α–2,3 glycans (3′SLN: Neu5Acα2-3Galβ1-4GlcNAcβ-SpNH-LC-LC-biotin) were kindly provided by the Consortium for Functional Glycomics (Scripps Research Institute, Department of Molecular Biology, La Jolla, CA, USA). Receptor-binding specificity was determined by a solid-phase direct binding assay as previously described (*Chandrasekaran et al., 2008*). Briefly, serial dilutions (0.195 µg/ml, 0.391 µg/ml, 0.781 µg/ml, 1.563 µg/ml, 3.125 µg/ml, 6.25 µg/ml, and 12.5 µg/ml) of biotinylated glycans 3′SLN and 6′SLN were prepared in PBS. Afterward, 100 µl was added to the wells of 96-well microtiter plates and allowed to attach overnight at 4 °C. The plates were then irradiated with 254 nm ultraviolet light for 2 min. After removing the glycopolymer solution, the plates were blocked with 0.1 ml of PBS containing 2% bovine serum albumin at room temperature for 1 hr. After washing with ice-cold PBS containing 0.1% Tween 20 (PBST), the plates were incubated in a solution containing influenza virus (64 HA units in PBST) at 4 °C for 12 hr. After washing with PBST, chicken antisera against Dk/KR/JS53/04 (avian H3N2), Cn/BJ/137/17 (canine H3N2), or BJ/1230/16 (human H3N2) virus were added to each well, and the plates were incubated at 4 °C for 2 hr. The wells were then washed with ice-cold PBST and incubated with HRP-linked goat anti-chicken antibody (Sigma-Aldrich) for 2 hr at 4 °C. After washing with ice-cold PBST, the plates were incubated with O-phenylenediamine in PBS containing 0.01% $H_2O_2$ for 10 min at room temperature. The reaction was stopped with 0.05 ml of 1 M $H_2SO_4$, and the absorbance was determined at 450 nm.

## Viral growth kinetics in cells

The $TCID_{50}$ was determined in MDCK cells by inoculation of 10-fold serially diluted viruses at 37 °C for 48 hr. The $TCID_{50}$ value was calculated by the Reed–Muench method. Multistep replication kinetics were determined using A549 and NHBE cells. A549 cells were infected with viruses at an MOI of 0.01, overlaid with serum-free DMEM containing 1 g/ml tosylsulfonyl phenylalanyl chloromethyl ketone (TPCK)-trypsin (Sigma-Aldrich) and incubated at 37 °C. NHBE cells were infected with viruses at an MOI of 0.01 and cultured in a B-ALI growth medium (Lonza) at 33 °C. The supernatants were sampled

at 2, 12, 24, 36, 48, 60, and 72 hpi and titrated by inoculating MDCK cells in 96-well plates. Three independent experiments were performed.

## Dog pathogenesis and transmission experiments

We used 10-week-old female beagles (Beijing Marshall Biotechnology Co., Ltd) seronegative for currently circulating influenza viruses as study subjects. The dogs were anesthetized with ketamine (20 mg/kg), xylazine (1 mg/kg), and infected intranasally with $10^{6.0}$ $EID_{50}$ of test virus in a 2 ml volume. The three animals of each group were subsequently euthanized at 4 dpi, and nasal turbinate, trachea, lung, and tonsil samples were collected for virus titration. Lung tissues were also used for pathological examination.

For transmission studies, groups of three female beagles housed in a cage placed inside an isolator were inoculated intranasally with $10^{6.0}$ $EID_{50}$ of the test virus. Twenty-four hours later, the three inoculated animals were individually paired by co-housing with a direct-contact dog. Nasal swabs were collected at 2 day intervals from 2 dpi to 14 dpi. Nasal secretion swabs were taken and placed in a 1.0 ml transmission medium [50% (vol/vol) glycerol in PBS] containing antibiotics. Viruses in the nasal secretion swabs were titrated in eggs. Sera were collected from contacted animals at 21 dpi. Seroconversion was analyzed by HI assay. Clinical signs and temperature were recorded daily for all inoculated dogs.

## Ferret pathogenesis and transmission experiments

Six-month-old female Angora ferrets (Wuxi Sangosho Biotechnology Co., Ltd, Angora) seronegative for currently circulating influenza viruses were used. In the transmission experiment, groups of three animals were each anesthetized with ketamine (20 mg/kg) and xylazine (1 mg/kg), and infected intranasally with $10^{6.0}$ $EID_{50}$ of test virus in a 500 µl volume (250 µl per nostril). Twenty-four hours later, the three inoculated animals were paired with naïve ferrets, housed in a wire-frame cage adjacent to the infected ferrets. The infected and respiratory droplet ferrets were 5 cm apart. Nasal washes were collected at 2 day intervals from 2dpi to 14dpi and titrated for viruses using eggs to monitor virus shedding. Sera were collected from respiratory droplet animals at 21 dpi. Seroconversion was analyzed by HI assay. The ambient conditions for these studies were 20 to 22°C and 30 to 40% relative humidity. The airflow in the isolator was horizontal at a speed of 0.1 m/s; the airflow direction was from the inoculated animals toward the exposed animals.

## HA thermostability

Purified viruses were diluted in PBS to 32 HA units and dispensed by 120 µl into 0.2 ml, thin-walled PCR tubes (USA Scientific, Ocala, FL). Tubes were placed into a Gradient Veriti 96-well thermal cycler (catalog number 9902; Life Technologies, Camarillo, CA). The temperature range was set at 48 to 63.0°C. Tubes were heated for 40 min and then transferred to ice 5 min. Control samples containing 120 µl of the virus were incubated for 40 min at 0 °C. The virus content in each sample was determined by an HA assay using a 1% suspension of chicken erythrocytes. Each virus sample was analyzed three times for thermostability.

## HA acid stability

HA acid stability was measured using syncytia assays. Vero cells in 24-well plates were infected at 3 PFU/cell MOI in the syncytia assay. At 16 hpi, infected cells were treated with DMEM supplemented with 5 mg/ml TPCK-treated trypsin for 15 min. Infected cells were subsequently maintained in pH-adjusted PBS buffers ranging from pH 5.2–5.5 or 5.6 for 15 min. After aspiration of pH-adjusted PBS, infected cells were incubated in DMEM supplemented with 5% FBS for 3 hr at 37 °C. The cells were then fixed and stained using Hema 3 Fixative and Solutions (Fisher Scientific). Photomicrographs of cells containing or lacking syncytia were recorded using a light microscope (*Russier et al., 2016*; *Reed et al., 2010*). A baseline of no virus-induced syncytia formation was obtained by exposing the cells to pH 5.5 media, under which Vero cells formed no visible syncytia. Micrographs were scored positive for syncytia formation if a field containing at least two syncytia that had at least five nuclei. HA activation pH values for syncytia assays were reported as the highest pH that induced syncytia as judged by positive scoring.

## Polymerase activity assay

A dual-luciferase reporter assay system (Promega, Madison, WI, USA) was used to compare the polymerase activities of viral RNP complexes (*Xu et al., 2016*). The indicated viruses' PB2, PB1, PA, and NP gene segments were separately cloned into the pCDNA3.1 expression plasmid. Mutations were introduced into the PB1 gene by site-directed mutagenesis (Invitrogen) according to the manufacturer's protocol. All constructs were sequenced to ensure the absence of unwanted mutations. The primer sequences used for cloning are available upon request. The PB2, PB1, PA, and NP plasmids (125 ng each plasmid) were used to transfect 293T cells with the Luci luciferase reporter plasmid (10 ng) and the renilla internal control plasmid (2.5 ng). Cultures were incubated at 33°C and 37°C. Cell lysates were analyzed 24 hr after transfection to measure firefly and renilla luciferase activities using GloMax 96 microplate luminometer (Promega).

## Statistical analyses

Differences between experimental groups were assessed using analysis of variance (ANOVA). $p < 0.05$ was considered to indicate a statistically significant difference.

## Nucleotide sequence accession numbers

The nucleotide sequences of 66 H3N2 CIVs isolated in this study are available in GenBank under accession numbers: ON877531-ON878058.

## Acknowledgements

This work was supported by the National Natural Science Foundation of China (32172838 and 32192451) and the 111 Project.

---

## Additional information

### Funding

| Funder | Grant reference number | Author |
|---|---|---|
| National Natural Science Foundation of China | Grant No. 32172838 | Yipeng Sun |
| National Natural Science Foundation of China | Grant No. 32192451 | Jinhua Liu |
| Higher Education Discipline Innovation Project | 111 Project | Yipeng Sun |

The funders had no role in study design, data collection and interpretation, or the decision to submit the work for publication.

### Author contributions

Mingyue Chen, Conceptualization, Data curation, Formal analysis, Validation, Investigation, Visualization, Methodology, Writing - original draft, Project administration, Writing - review and editing; Yanli Lyu, Conceptualization, Resources, Data curation, Supervision, Investigation, Methodology, Project administration; Fan Wu, Conceptualization, Data curation, Validation, Investigation, Visualization, Methodology; Ying Zhang, Resources, Data curation, Investigation, Methodology; Hongkui Li, Conceptualization, Resources, Data curation, Investigation; Rui Wang, Data curation, Validation, Investigation, Visualization; Yang Liu, Resources, Data curation, Supervision, Investigation, Methodology; Xinyu Yang, Validation, Investigation, Visualization, Methodology; Liwei Zhou, Resources, Data curation, Validation, Investigation, Visualization, Methodology; Ming Zhang, Conceptualization, Resources, Formal analysis, Methodology, Project administration; Qi Tong, Resources, Supervision, Investigation, Methodology, Project administration; Honglei Sun, Conceptualization, Resources, Data curation, Formal analysis, Supervision, Project administration, Writing - review and editing; Juan Pu, Conceptualization, Resources, Data curation, Formal analysis, Project administration; Jinhua Liu,

Conceptualization, Resources, Formal analysis, Supervision, Project administration, Writing - review and editing; Yipeng Sun, Conceptualization, Resources, Formal analysis, Supervision, Methodology, Project administration, Writing - review and editing

### Author ORCIDs
Yipeng Sun (iD) http://orcid.org/0000-0001-9399-0039

### Ethics
All experiments with live viruses were performed in animal biosafety level 2 (ABSL-2) environments. The present study was carried out in accordance with the Guide for the Care and Use of Laboratory Animals of the Ministry of Science and Technology of the People's Republic of China. The protocols for the animal studies were approved by the Committee on the Ethics of Laboratory Animals of China Agricultural University (approval SKLAB-B-2010-003).

### Decision letter and Author response
Decision letter https://doi.org/10.7554/eLife.83470.sa1
Author response https://doi.org/10.7554/eLife.83470.sa2

---

## Additional files

### Supplementary files
• Supplementary file 1. Antigenic analysis of H3N2 subtype canine influenza viruses in the world. HI titers are the inverse of the highest dilution that inhibited hemagglutination. Cells containing moderate HI titers (160, 320) are shaded gray, and high titers (>640) black. Low HI titers (40, 80) are not shaded. Titers <10 are indicated with the sign <. HI, hemagglutinin inhibition. †Homologous titer.

• Supplementary file 2. NI titers of ferret antisera against human influenza A (H3N2) and H3N2 canine influenza preponderant prevalent viruses used in this study. NI titers are the inverse of the highest dilution that inhibited neuraminidase activity. NI, neuraminidase inhibition. †Homologous titer.

• Supplementary file 3. Seroconversion of the dogs in transmission experiments. ªSera were collected from the dogs 3 weeks after virus inoculation or exposure; these animals were used for the transmission studies shown in *Figure 4* and *Figure 4—figure supplement 1*. HI, hemagglutinin inhibition.

• Supplementary file 4. Seroconversion of the ferrets in transmission experiments. ªSera were collected from the ferrets 3 weeks after virus inoculation or exposure; these animals were used for the transmission studies shown in *Figure 5*. HI, hemagglutinin inhibition.

• Supplementary file 5. Transmission of H3N2 reassortant viruses in dogs. ‡Number of animals in which virus was detected in nasal swabs. Values in parentheses indicate nasal swab titers, which are expressed as mean $log_{10}$ peak virus titer observed within the first 9 dpi. §Number of animals in which seroconversion was observed. Values in parentheses show HI antibody titers in sera collected at 21 dpi. HI titers were determined using the homologous virus as a test antigen.

• Supplementary file 6. Transmission of H3N2 reassortant viruses in ferrets. ‡Number of animals in which virus was detected in nasal washes. Values in parentheses indicate nasal wash titers, which are expressed as mean $log_{10}$ peak virus titer observed within the first 9 dpi. §Number of animals in which seroconversion was observed. Values in parentheses show HI antibody titers in sera collected at 21 dpi. HI titers were determined using the homologous virus as a test antigen.

• MDAR checklist

### Data availability
Sequencing data have been deposited in GenBank under accession codes ON877531-ON878058.

The following dataset was generated:

| Author(s) | Year | Dataset title | Dataset URL | Database and Identifier |
|---|---|---|---|---|
| Sun Y, Chen M | 2022 | Primer sequences | https://doi.org/10.5061/dryad.4qrfj6qdf | Dryad Digital Repository, 10.5061/dryad.4qrfj6qdf |

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
