## [Editor Report]

This paper focuses on the avian H3N2 influenza virus that has recently started infecting and spreading between dogs. Using exhaustive and impressive experimental approaches, the authors demonstrate how this virus is adapting to dogs over time, gaining more and more properties consistent with robust infection of mammals. This paper is destined to become part of the canon on emerging viruses.

---

## [Decision Letter]

**Decision letter after peer review:**

Thank you for submitting your article "Increased public health threat of avian-origin H3N2 influenza virus during evolution in dogs" for consideration by *eLife*. Your article has been reviewed by 3 peer reviewers, one of whom is a member of our Board of Reviewing Editors, and the evaluation has been overseen by Sara Sawyer as the Senior Editor. The following individual involved in the review of your submission has agreed to reveal their identity: Hongquan Wan (Reviewer #1).

The reviewers have discussed their reviews with one another, and the Reviewing Editor has drafted this letter to help you prepare a revised submission. Personally, I am very excited about this paper, and I hope you will be able to address the outstanding issues.

Essential revisions:

1) Anyone who reads this paper is going to wonder if human seasonal H3N2 will provide us protection from this virus if it does jump from dogs to humans. The authors indicate not, but they state this is a single sentence in the paper and a supplemental table. This fact needs to be emphasized (possibly even in the abstract). Data needs to be brought into the main paper, and also anti-NA antibody assays must be performed. (One of reasons proposed as to the lower impact of the H3N2 pandemic was the presence of anti-N2 antibodies in the human population.). Regardless of the outcome of the N2 experiment, we would still be interested in the data being in the main paper. Either way the paper is still very compelling.

2) Please work with a professional English editor to strengthen the writing of the paper.

*Reviewer #1 (Recommendations for the authors):*

1. Please improve the writing of the manuscript.

2. Page 3 line 65: "…the Eurasian avian-origin lineage that originated in …". Which subtype(s) of influenza virus?

3. Page 4 lines 91-92: "…and has spread to and circulated in the United States since 2015…" Any evidence that the US canine H3N2 viruses were from China and South Korea? If not, please modify this sentence.

4. Page 8 line 161: "…4-to more-fold lower than…" It might be better to write as "{greater than or equal to} 4-fold lower than". Lines 166-167: "The co-circulation of different antigenic group viruses in recent years increased the difficulty of preventing and controlling canine influenza viruses." Any control strategy implemented for controlling canine influenza viruses in China and/or other countries?

5. Page 9 lines 182-184: "These results indicated that preexisting immunity derived from the present human seasonal influenza viruses cannot provide protection against H3N2 CIVs." This statement is not correct. It might be not that the preexisting immunity derived from the present human seasonal influenza viruses can't provide production, based on your serological data it is rather that a very small proportion of the cohort from whom you collected sera have immunity against these viruses.

6. Page 10 line 199: change "Cn/Hain/079/19" to Cn/HaiN/079/19.

7. Pages 13-16 lines 238-275. It is hard to follow this section. Line 240: "…inoculated intranasally six dogs with 10^6^ TCID50 of each virus strain." The authors are wished to either specify the virus strains here or cite figures so that the readers can know the viruses that were used in the infection of dogs.

Lines 243-245: "…H3N2 CIVs (from clades 0, 1, 2, 3, and 4) caused only mild clinical signs with mean clinical scores ranging from 0.5 to 2.5 and a mean body temperature ranging from 37.7°C to 39.7°C (Figure 4A and 4B)." The authors should cite Figure S6 as Figures 4A and 4B did not show data for clades 0 and 1 viruses. It should also be specified how many animals in each group were followed up for clinical signs. In this reviewer's opinion, if the dogs sacrificed for virus titration were not included for the monitoring of clinical signs, the authors should describe the virus titration first, and then the clinical signs and virus transmission. To do this, Figure 4C can be adjusted to be Figure 4A and the original Figures 4A and 4B can then be shown as Figures 4B and 4C.

For the legend to Figure 4, the number (n) of animals shown in each panel should be specified. Lines 272-273: "Direct contact transmission of H3N2 CIVs in dogs." Instead of such a very general title, the authors are wished to write in a bit detail specifying what were shown in panel D to demonstrate the contact transmission.

8. Page 16 line 288: change "Cn/US/BRW003/18" to Cn/CA/BRW003/18 and "at" should be changed to "by".

9. Page 22 line 399: "Therefore, we evaluated the potential threat of H3N2 CIVs…" Threat to what?

10. Page 23 lines 425-426: "…in nine provinces of China (Liaoning, Beijing, Tianjin, Nanjing, Shanghai, Shanxi, Guangzhou, Fujian, and Hainan)." Please modify this sentence as Beijing, Tianjin, Nanjing, Shanghai and Guangzhou are cities/municipalities but not provinces.

11. Page 26 lines 490-491: "The human sera were selected from children, adults, and elderly adults…" When and where were these sera collected? Any information about the history of influenza vaccination of the subjects?

*Reviewer #2 (Recommendations for the authors):*

My only major comment is that a professional English language editor is desperately needed. The data presentation in figures is excellent, but the writing is problematic throughout. *eLife* might want to provide some assistance on making the abstract as strong as possible.

*Reviewer #3 (Recommendations for the authors):*

Anti-NA antibody assays must be performed.

---

## [Author Response]

Reviewer #1 (Recommendations for the authors):1. Please improve the writing of the manuscript.

We commissioned a language editing company to improve our manuscript.

2. Page 3 line 65: "…the Eurasian avian-origin lineage that originated in …". Which subtype(s) of influenza virus?

The subtype of influenza virus has been added (line 60).

3. Page 4 lines 91-92: "…and has spread to and circulated in the United States since 2015…" Any evidence that the US canine H3N2 viruses were from China and South Korea? If not, please modify this sentence.

This statement might not be very exact because there is no evidence that H3N2 CIVs in the United States is directly transported from South Korea or China, and thus we modified this sentence (line 87).

4. Page 8 line 161: "…4-to more-fold lower than…" It might be better to write as "{greater than or equal to} 4-fold lower than".

We changed "…4-to more-fold lower than…" to "greater than or equal to four-fold lower than" as suggested (line 159).

Lines 166-167: "The co-circulation of different antigenic group viruses in recent years increased the difficulty of preventing and controlling canine influenza viruses." Any control strategy implemented for controlling canine influenza viruses in China and/or other countries?

Commercially available inactivated H3N2 CIV and H3N2/H3N8 CIV vaccines have been approved and used in the United States since 2015. To our knowledge, several corporations in China are developing H3N2 CIV vaccines.

5. Page 9 lines 182-184: "These results indicated that preexisting immunity derived from the present human seasonal influenza viruses cannot provide protection against H3N2 CIVs." This statement is not correct. It might be not that the preexisting immunity derived from the present human seasonal influenza viruses can't provide production, based on your serological data it is rather that a very small proportion of the cohort from whom you collected sera have immunity against these viruses.

We agree with the reviewer. To make the conclusion more rigorous, we revised this sentence. This sentence is not only a summary of the serological data but also of the whole section. We moved this sentence to a separate paragraph (lines 187-189).

6. Page 10 line 199: change "Cn/Hain/079/19" to Cn/HaiN/079/19.

It has been revised as suggested (line 209).

7. Pages 13-16 lines 238-275. It is hard to follow this section. Line 240: "…inoculated intranasally six dogs with 10^6^ TCID50 of each virus strain." The authors are wished to either specify the virus strains here or cite figures so that the readers can know the viruses that were used in the infection of dogs.

We cited Figure 4—figure supplement 1 in accordance with the comments (lines 251-252).

Lines 243-245: "…H3N2 CIVs (from clades 0, 1, 2, 3, and 4) caused only mild clinical signs with mean clinical scores ranging from 0.5 to 2.5 and a mean body temperature ranging from 37.7°C to 39.7°C (Figure 4A and 4B)." The authors should cite Figure S6 as Figures 4A and 4B did not show data for clades 0 and 1 viruses.

We cited Figure 4—figure supplement 1 (Figure S6) (line 264).

It should also be specified how many animals in each group were followed up for clinical signs.

The clinical signs of three inoculated dogs used for the transmission experiment were monitored (lines 288-289). We added this information to the manuscript (lines 260-261).

In this reviewer's opinion, if the dogs sacrificed for virus titration were not included for the monitoring of clinical signs, the authors should describe the virus titration first, and then the clinical signs and virus transmission. To do this, Figure 4C can be adjusted to be Figure 4A and the original Figures 4A and 4B can then be shown as Figures 4B and 4C.

The dogs sacrificed for virus titration were not included in the monitoring of clinical signs. As the reviewer suggested, we adjusted the order of the related description and figures (lines 253-259).

For the legend to Figure 4, the number (n) of animals shown in each panel should be specified. Lines 272-273: "Direct contact transmission of H3N2 CIVs in dogs." Instead of such a very general title, the authors are wished to write in a bit detail specifying what were shown in panel D to demonstrate the contact transmission.

The number of animals shown in each panel has been added. Panel D has been described in detail as suggested (lines 289-292).

8. Page 16 line 288: change "Cn/US/BRW003/18" to Cn/CA/BRW003/18 and "at" should be changed to "by".

They have been changed as suggested (line 307).

9. Page 22 line 399: "Therefore, we evaluated the potential threat of H3N2 CIVs…" Threat to what?

Threat to public health. We corrected this sentence (line 421).

10. Page 23 lines 425-426: "…in nine provinces of China (Liaoning, Beijing, Tianjin, Nanjing, Shanghai, Shanxi, Guangzhou, Fujian, and Hainan)." Please modify this sentence as Beijing, Tianjin, Nanjing, Shanghai and Guangzhou are cities/municipalities but not provinces.

We modified this sentence in accordance with the comments (lines 452-453).

11. Page 26 lines 490-491: "The human sera were selected from children, adults, and elderly adults…" When and where were these sera collected? Any information about the history of influenza vaccination of the subjects?

These sera were collected in 2021 from the First Medical Centre of Chinese PLA General Hospital in Beijing. The information on the influenza vaccination history of these samples was unknown. We added the related information in the manuscript (lines 521-523).

Reviewer #3 (Recommendations for the authors):Anti-NA antibody assays must be performed.

We performed neuraminidase inhibition assays as suggested for both ferret sera against human H3N2 virus and human sera. The results showed that the NI titers of ferret sera against human H3N2 virus to canine H3N2 viruses were <10 (lines 168- 169, Supplementary file 2). Additionally, 2.0%–3.0% of the children's serum samples, 1.0%–2.0% of the adult's serum samples, and 1.0%–2.0% of the elderly adult's serum samples had NI antibody titers of ≥10 to canine origin NA (lines 179-182, Table 1, and lines 529-539).